# Impact of adherence to a lifestyle-integrated programme on physical function and behavioural complexity in young older adults at risk of functional decline: a multicentre RCT secondary analysis

A Stefanie Mikolaizak [1], Kristin Taraldsen [2,3], Elisabeth Boulton [4,5,6], Katharina Gordt [1], Andrea Britta Maier [7], Sabato Mellone [8], Helen Hawley-Hague [4,6], Kamiar Aminian [9], Lorenzo Chiari [8], Anisoara Paraschiv-Ionescu [9], Mirjam Pijnappels [10], Chris Todd [4,6,11], Beatrix Vereijken [3], Jorunn L Helbostad [3], Clemens Becker [1]

For numbered affiliations see end of article.

**Correspondence to**
Dr A Stefanie Mikolaizak;
Stefanie.Mikolaizak@rbk.de

## ABSTRACT

**Context** Long-term adherence to physical activity (PA) interventions is challenging. The Lifestyle-integrated Functional Exercise programmes were adapted Lifestyle-integrated Functional Exercise (aLiFE) to include more challenging activities and a behavioural change framework, and then enhanced Lifestyle-integrated Functional Exercise (eLiFE) to be delivered using smartphones and smartwatches.

**Objectives** To (1) compare adherence measures, (2) identify determinants of adherence and (3) assess the impact on outcome measures of a lifestyle-integrated programme.

**Design, setting and participants** A multicentre, feasibility randomised controlled trial including participants aged 61–70 years conducted in three European cities.

**Interventions** Six-month trainer-supported aLiFE or eLiFE compared with a control group, which received written PA advice.

**Outcome measures** Self-reporting adherence per month using a single question and after 6-month intervention using the Exercise Adherence Rating Scale (EARS, score range 6–24). Treatment outcomes included function and disability scores (measured using the Late-Life Function and Disability Index) and sensor-derived physical behaviour complexity measure. Determinants of adherence (EARS score) were identified using linear multivariate analysis. Linear regression estimated the association of adherence on treatment outcome.

**Results** We included 120 participants randomised to the intervention groups (aLiFE/eLiFE) (66.3±2.3 years, 53% women). The 106 participants reassessed after 6 months had a mean EARS score of 16.0±5.1. Better adherence was associated with lower number of medications taken, lower depression and lower risk of functional decline. We estimated adherence to significantly increase basic lower extremity function by 1.3 points (p<0.0001), advanced lower extremity function by 1.0 point (p<0.0001) and

behavioural complexity by 0.008 per 1.0 point higher EARS score (F(3,91)=3.55, p=0.017) regardless of group allocation.

**Conclusion** PA adherence was associated with better lower extremity function and physical behavioural complexity. Barriers to adherence should be addressed preintervention to enhance intervention efficacy. Further research is needed to unravel the impact of behaviour change techniques embedded into technology-delivered activity interventions on adherence.

**Trial registration number** NCT03065088.

## Strengths and limitations of this study

⇒ Our study includes data from a three-arm, three-site, international research study, thus eliminating cultural bias towards the intervention.
⇒ Adherence was defined a priori better to capture intervention impact.
⇒ A complier average causal effect analysis was not possible; therefore, association between adherence and outcome could only be established, not causation.
⇒ A comprehensive collection of adherence data was included, both as a single question reported monthly throughout the study period as well as a questionnaire at the assessments at 6 and 12 months.
⇒ The monthly measurement of adherence may have acted as a prompt to complete planned activities, although this would be common across all intervention groups and participants.

## BACKGROUND

Exercise interventions conducted as randomised controlled trials (RCTs) are

considered the gold standard for allowing causal inference to be made regarding the efficacy of a treatment. However, the reality is that the participants' adherence to treatment is typically imperfect. The actual intervention effect can only be evaluated accurately if adherence levels are high. Thus, conclusions on treatment effects of an exercise intervention will only be valid if the adherence levels to the protocol are evaluated,[1 2] with several studies reporting poor adherence rates among the study population to mediate the lack of treatment effect.[3] Further disentangling treatment (eg, type of delivery mode) and exercise effect (eg, type of exercise) on outcomes is essential when promoting benefits achieved through exercise interventions.

The World Health Organization (WHO) defines adherence as the 'extent to which a person's behaviour corresponds with the recommendations from a healthcare provider',[4] and it involves an active choice from the participants to follow through and to take responsibility for their well-being. In pharmaceutical trials, the intervention is 'unidimensional' and is based on patients taking the required dose, which can be monitored using bioanalytical methods. Interventions involving exercise and physical activity (PA) are more complex and 'multidimensional', as, for example, variance in daily form/fitness or required training effect influences the intensity required to maintain a specific training dose.

To date, there is no consensus regarding which adherence measurement tool should be used, nor when and how adherence should be measured.[5–7] Besides the difficulty of recording adherence rates, achieving sufficiently high adherence is challenging. High adherence is frequently documented in the early stages of an exercise intervention, regardless of delivery mode and exact definition used.[2 3] In contrast, long-term intervention adherence can be as low as 30%,[2] with activity levels reverting to preintervention levels following the intervention period. Integrating tailored activities into daily life helps achieve longer-term adherence levels than a structured programme, where exercise is segregated from daily routines.[8] While this delivery method is gaining in popularity,[9] there is limited knowledge on how to assess adherence to a lifestyle-integrated activity programme. Furthermore, a recent systematic review showed that interventions that used telecommunication, as well as those offering lifestyle-integrated exercise interventions, have appeared most promising for maintaining adherence.[3] Despite this, interventions that specifically targeted adherence do not affect balance or gait parameters.

Adherence to exercise appears to be influenced by socioeconomic status, health, physical function, motivation and other psychological variables.[10 11] Understanding which factors influence young older adults (60–70 years old) to adhere to lifestyle-integrated exercises is essential for future intervention success.[3] Modifiable factors that influence adherence need to be identified, ideally pretreatment, in order to influence treatment effect.

This paper reports *a priori* secondary analysis of the PreventIT randomised controlled feasibility trial data to investigate factors influencing adherence to a personalised lifestyle-integrated exercise intervention offered to young older adults, with the intention to prevent accelerated functional decline. Here, adherence is defined as participants completing all their planned lifestyle-integrated activities into daily life, as described further. Further, adherence was hypothesised to be a treatment-effect modifier which needs to be analysed within the intervention groups. The current, a priori analysis, based on adherence levels is thus warranted. It may have greater clinical relevance, be more indicative of the provided treatment effect,[12] aid future improvements of the intervention programme and guide implementation into practice.

## Aims

The aims of this analysis were to (1) assess whether measuring adherence monthly prospectively, and retrospectively every 6 months, yields the same results; (2) identify baseline characteristics that determine adherence levels; and (3) estimate the association of adherence to a lifestyle-integrated intervention on primary and secondary outcome measures (functional ability and behavioural complexity) in young older adults.

## METHODS

### Design

This a priori analysis is part of a large-scale multicentre RCT, PreventIT (Clinical trial NCT03065088), analysing the impact of a 6-month active intervention period within a 12-month trial. PreventIT, a personalised behavioural change exercise intervention for young older adults, was developed to prevent accelerated functional decline. The detailed protocol[13] and results of the feasibility analysis, including an estimate of change for the primary outcome measures, have been published elsewhere.[14]

### Participants

The trial sample for this preplanned analysis includes the intervention group participants from the two treatment modes (n=120). Participants were invited to participate if they were aged 61–70 years, retired or working part-time, community dwelling and able to walk 500 m without a walking aid. Exclusion criteria included participation in an exercise class (>1/week) or undertaking moderate-intensity PA (≥150 min/week).

### Intervention

Participants received tailored exercises at an individual level and learnt to integrate these into everyday situations. The same activity framework was applied in both intervention arms and was delivered in two different modes: via paper–pen manuals (adapted Lifestyle-integrated Functional Exercise (aLiFE)) or via a mobile health application system (enhanced Lifestyle-integrated Functional

Exercise (eLiFE)). Participants received between one and six home visits, depending on group allocation, from an exercise physiologist/scientist or physiotherapist across 6 months during which the exercise programme was taught and supported, with the intention for participants to learn to progress their exercises independently. The programme consisted of strategies to (1) improve balance, (2) increase muscle strength, and (3) reduce sedentariness and increase PA. The concept, based on the Lifestyle-integrated Functional Exercise (LiFE) programme,[8] has been adapted to young older adults to be more challenging[15] and is underpinned by a behavioural change framework to support older adults to form long-term PA habits. PreventIT has taken the original LiFE concept and further developed the behaviour change elements, explicitly mapping them to social cognitive theory, habit formation theory, and 30 behaviour change techniques (BCTs).[16] Goal setting, planning, prompts and real-time feedback are used to deliver a person-centred experience. The full interventions details have been published.[13]

Participants received their randomly allocated intervention during the first 6 months (active period) of the programme and were then encouraged to keep up their programme (passive period) independently for another 6 months. Here, an analysis of the first 6-month active intervention period is presented.

For context, control group participants received general WHO advice on the benefits of PA but no tailored exercise programme. Therefore, an exercise adherence analysis was not relevant for this group.

Exercise adherence was encouraged in general for all participants, and the participants were provided with adherence logs to document their activity. No incentives or additional adherence promotion was provided to the participants within the PreventIT trial.

### Outcome measures
Participants underwent a comprehensive assessment by an assessor blinded to group allocation at baseline, after a 6-month active and a further 6-month passive intervention period.

### Adherence measures
Adherence was collected using two methods in this trial.

### *Exercise Adherence Rating Scale (EARS) at post-test and follow-up*
The EARS[17] was completed by participants during follow-up as an objective, reliable and validated self-report measure of adherence to planned (exercise) activities. EARS consists of 16 questions, each answered on a 5-point Likert-scale (scored 0–4), identifying how strongly the participant agrees/disagrees with each statement. Six questions capture the level of adherence, while 10 questions address facilitators or barriers of adherence. A higher score indicates better adherence; however, a predefined cut-off point for adequate adherence does not exist. Adherence is frequently dichotomised, based on an arbitrary cut-off; however, adherence with an ongoing complex treatment is variable, and analysing adherence as a continuous or ordinal variable is warranted.[18]

### *Monthly reports of adherence*
Every month, participants were asked to report on a single-page questionnaire by ticking one of seven options, whether they had completed their planned activities 'yes—more than planned', 'yes—as much as planned', 'yes— but not as much as planned' or 'no—not at all' because (1) 'they did not feel well', (2) 'they forgot', (3) 'they did not like the activities' or (4) 'they did not have time'. Response to the single question was possible via email or postcard, where each month of reporting was summarised as full adherence (responded positively and 'more than or as much as planned'), partial adherence (responded positively and 'but not as much as planned') or non-adherence (not at all, regardless of reason). Data from participants who provided four or more responses during the 6-month intervention and follow-up period were included in the analysis. The validity of this novel method of assessing adherence was tested within the PreventIT feasibility trial.[14]

### Treatment outcome measures
The trial included two primary outcome measures[13]: subjective health rating using the Late-Life Function and Disability Index (LLFDI)[19] and PA measure using a behavioural complexity metric.[20]

The LLFDI is a comprehensive questionnaire assessing function (ability to perform specific activities of daily living) and disability (inability to take part in major life tasks and social roles) for use in community-dwelling older adults.[19 21]

The behavioural complexity metric was derived from a 7-day consecutive PA monitoring period and assessed in the domains of daily PA and social participation. A three-axis logging accelerometer (https://axivity.com/product/ax3) attached to participants' lower back (using adhesive tape) represented physical behaviours as time series embedding activity characteristics (ie, type, duration, intensity and dynamics of transitions between activities).[20 22] The concept of behavioural complexity, initially developed to assess physical behaviour in patients with chronic pain,[22 23] was further developed in the PreventIT trial using novel computational complexity methods and the relationship of complexity metrics with additional clinical scores/outcomes, such as fear of falling, and functional balance and mobility performance preintervention and postintervention.[20 24] The value of behavioural complexity metric ranges from a minimal value of 0.1 for very impaired frail older people to a maximal value of around 0.7 for highly active subjects. For context, published data for well-functioning older adults indicated a value of 0.40±0.07 (mean±SD) for fully confident older adults without fear of falling and 0.30±0.06 for those active but less confident in their ability.[20]

Additional outcome measures included general health and function, medication use, neuropsychological

**Table 1** Demographics

| | aLiFE<br>n=59 | eLiFE<br>n=61 |
|---|---|---|
| Age (years), mean (SD) | 66.19 (2.32) | 66.43 (2.33) |
| Gender (female), n (%) | 30 (50.8) | 33 (54.1) |
| Living alone, n (%) | 21 (36.7) | 18 (29.5) |
| Pain during rest (0–10), median (IQR) | 1.0 (1.0–3.0) | 1.0 (1.0–3.0) |
| Pain during walking (0–10), mean (SD) | 2.5 (1.0–4.0) | 2.0 (1.0–4.5) |
| Falls in past year, n (%) | | |
| 0 | 53 (91.4) | 51 (83.6) |
| 1 | 4 (6.9) | 10 (16.4) |
| 2+ | 1 (1.7) | 0 (0) |
| Economic satisfaction, n (%) | | |
| Good | 23 (39.7) | 23 (37.7) |
| Sufficient | 22 (37.9) | 31 (50.8) |
| Poor/bad | 13 (22.4) | 7 (11.5) |
| Total number comorbidities, median (IQR) | 2.0 (1.0–4.0) | 2.0 (1.0–3.0) |
| Total number medications, median (IQR) | 2.0 (1.0–4.0) | 2.0 (1.0–3.0) |
| Confirmed arthritis, n (%) | 19 (32.8) | 18 (29.5) |
| Confirmed cardiovascular disease, n (%) | 9 (15.5) | 14 (23.0) |
| CES-D score (0–60), median (IQR) | 8.0 (3.0–14.0) | 6.5 (3.25–11.0) |
| Moderate risk of functional decline, n (%) | 7 (11.9) | 6 (10) |
| LLFDI score (0–100), mean (SD) | | |
| Functional (overall) | 73.7 (12.9) | 73.1 (10.6) |
| Lower extremities basic | 85.3 (15.3) | 83.6 (12.7) |
| Lower extremities advanced | 70.5 (15.7) | 71.5 (15.4) |
| Upper extremities | 88.0 (12.7) | 87.7 (12.1) |
| Disability frequency | 52.2 (4.6) | 50.5 (4.0) |
| Disability limitation (0–100), median (IQR) | 84 (72.6–100.0) | 80 (72–100) |
| Behavioural complexity score, mean (SD) | 0.347 (0.123) | 0.374 (0.119) |

aLiFE, adapted Lifestyle-integrated Functional Exercise; CES-D, Center for Epidemiological Studies Depression; eLiFE, enhanced Lifestyle-integrated Functional Exercise; LLFDI, Late-Life Function and Disability Index.

assessment, physical function and risk of functional decline (table 1). The PreventIT risk screening tool was used to identify and provide a participant's risk estimate (low, medium or high risk) for functional decline over the next 9 years.[25]

### Determinants of adherence

Key factors which were anticipated to be positively or negatively associated with participants' adherence were selected from the outcome measures obtained during the PreventIT baseline assessment.[14] Determinates were chosen if they were (1) known functional decline risk factors (eg, age or number of medications);[26 27] (2) strength or balance deficits, identified during assessment;[6] (3) potentially associated with adherence (eg, pain when walking and cognitive impairment).[3–7 13]

### Statistical analysis

Descriptive statistics were used to summarise demographic data. Baseline comparisons depending on adherence levels were performed using parametric and non-parametric tests as appropriate. Initially, the association between adherence, as reported monthly during the active intervention period (0–6 months postrandomisation), and EARS sum score of questions 1–6, reported at 6 months post randomisation, was assessed using Pearson's correlation. In further analyses, the continuous EARS score was used.

The modal response was used as an imputation method to account for missing monthly reporting data, which was missing completely at random, allowing 106 participants (53 from each treatment arm) to be included in univariate/multivariate analysis to reduce bias.[28]

Univariate associations between potential determinants and adherence (EARS score) were assessed using logistic regression. To ensure the collinearity assumption was met, potential variables were required to measure/assess uniquely different outcomes, and a restricted number of variables were selected, determined by the number of available observations to reduce the risk of type I error. Variables identified during univariate regression (p<0.05) were subsequently entered into a backward (elimination) multivariate linear regression model to determine the best set of explanatory variables.

In lieu of a suitable complier average causal effect (CACE) analysis, which was not possible given the intention-to-treat analysis showed no difference in effect between groups, multiple linear regression was performed while still honouring the CACE assumptions.[29] Multiple linear regression was used to estimate the association of predicted adherence on primary and secondary outcome measures at 6 months of follow-up, using group allocation (aLiFE and eLiFE), baseline values and predicted adherence levels as independent variables. Participants' predicted adherence levels (EARS score) were derived using multiple linear regression results.[30 31] Predicted adherence levels were centred,[32] and the interaction association between group allocation and adherence level is presented only when significant. Main associations of baseline values, group allocation and adherence levels are presented. Analyses were conducted using SPSS V.25.0.

## RESULTS

Fifty-nine participants were randomised to receive the traditional paper–pen based (aLiFE) and 61 were to receive the technology supported (eLiFE) programme. The 6-month follow-up was completed by 53 (89.8%) aLIFE and 53 (86.9%) eLIFE participants. Participant characteristics are presented in table 1. The mean age of the participants was 66 years (SD 2.3), and an equal number of men and women were enrolled. Most participants had not experienced a fall in the past year, were taking two medications daily and were overall in good health.

### Monthly and 6-month adherence measurement

During the 6-month follow-up period, which also included monthly follow-up, 26 (24.5%) participants reported fully adhering to their planned activity, while 21 (19.8%) participants reported non-adherence (table 2). The remainder (59, 55.7%) were classified as partial adherers. During the 6-month follow-up assessment, the mean EARS score was 16.02 (SD 5.12). These results showed a large, positive and significant association between the monthly adherence reporting and the EARS questionnaire score at 6 months (r(102)= 0.618, p<0.001). The mean EARS scores increased with increased adherence, as measured on a monthly basis, with non-adherers scoring as low as 4 out of 24 points and full adherers having a mean of 20 out of 24 on their EARS score.

### Determinants of adherence

Univariate analyses identified nine baseline variables to be associated with intervention adherence using EARS (table 3), of which three variables were retained in the multivariate linear regression model: greater number of medication (beta=−0.21, p=003), CES-D score (beta=−0.25, p=0.009) and the risk of functional decline score (beta=−0.21, p=0.35) all significantly influenced poorer adherence levels.

### Estimated association of adherence on outcome measures

After controlling for group allocation, predicted adherence level was positively associated with improved physical function, behavioural complexity and well-being (table 4) at follow-up. Group allocation was not significantly associated with an improvement in any primary or secondary outcome measure.

**Table 2** Reported adherence levels

| Adherence level | | Monthly adherence reporting | | | | EARS score at 6 months of follow-up (/24), mean (SD) (n=106) |
|---|---|---|---|---|---|---|
| | | Full adherence | Partial adherence | Non-adherence | Missing | |
| Treatment mode | | | | | | |
| All | | 26 | 59 | 21 | 14 | 16.02 (5.12) |
| eLiFE | | 15 | 27 | 10 | 8 | 16.75 (5.01) |
| aLiFE | | 11 | 32 | 11 | *6* | 15.29 (5.18) |
| EARS score, mean (min, max) | All | 20.3 (12, 24) | 15.6 (7, 24) | 14.1 (4, 22) | | |
| | eLIFE | 20.6 (12, 24) | 16.0 (7, 24) | 16.0 (6, 14) | | |
| | aLIFE | 19.9 (16, 24) | 15.3 (8, 24) | 12.2 (4, 22) | | |

Missing: 3 participants failed to report their monthly adherence repeatedly, beyond where imputation was possible; further 11 withdrew during follow-up.
aLIFE, adapted Lifestyle-integrated Functional Exercise; EARS, Exercise Adherence Rating Scale; eLife, enhanced Lifestyle-integrated Functional Exercise.

**Table 3** Determinants of adherence level (measured using the Exercise Adherence Rating Scale)

| | Univariate | | | Multivariate* | | |
|---|---|---|---|---|---|---|
| | B (SE) | ß | P value | B (SE) | ß | P value |
| **Sociodemographic** | | | | | | |
| Gender (female) | 1.56 (1.01) | 0.14 | 0.153 | | | |
| Age (years)† | −0.12 (0.22) | −0.05 | 0.587 | | | |
| Satisfaction with economic situation† | −0.01 (0.72) | −0.002 | 0.987 | | | |
| Living status (alone)‡ | 2.12 (1.06) | 0.20 | **0.048** | | | |
| Years of education‡ | 0.003 (0.11) | 0.003 | 0.978 | | | |
| **Medical history** | | | | | | |
| Number of medications† | −0.61 (0.21) | −0.29 | **0.003** | −0.45 (0.21) | −0.21 | **0.033** |
| Diagnosed/treated for arthritis§ | −2.10 (1.08) | −0.19 | 0.056 | | | |
| Diagnosed/treated for cardiovascular diseases§ | −0.47 (1.30) | −0.04 | 0.721 | | | |
| **Cognition, affect and well-being** | | | | | | |
| MOCA score‡ | 0.27 (0.27) | 0.10 | 0.337 | | | |
| CES-D score‡ | −0.20 (0.08) | −0.25 | **0.011** | −0.20 (0.07) | −0.25 | **0.009** |
| Short Form 12 score‡ | 0.48 (0.17) | 0.28 | **0.006** | | | |
| **Physical ability and mobility§** | | | | | | |
| Gait speed, usual pace (4 m walk) | 5.56 (2.29) | 0.24 | **0.017** | | | |
| Gait speed, usual pace (7 m walk) | 4.30 (2.02) | 0.21 | **0.036** | | | |
| Fast gait speed, fast pace (7 m walk) | 1.82 (1.02) | 0.18 | 0.077 | | | |
| CBMS (/96) | 0.11 (0.04) | 0.28 | **0.004** | | | |
| 8-Level Balance test (/8)§ | 0.93 (0.52) | 0.18 | 0.077 | | | |
| Cadence, mean (SD) | 0.07 (0.11) | 0.07 | 0.517 | | | |
| Complexity, mean (SD) | −0.10 (4.13) | −0.002 | 0.981 | | | |
| **Subjective health rating** | | | | | | |
| Pain while walking, VAS‡ | −0.31 (0.22) | −0.14 | 0.151 | | | |
| Pain during rest, VAS ‡ | −0.03 (0.26) | −0.01 | 0.915 | | | |
| LLFDI disability frequency (/100) | 0.17 (0.12) | 0.14 | 0.150 | | | |
| LLFDI disability limitation (/100) | 0.08 (0.04) | 0.21 | **0.032** | | | |
| LLFDI function total (/100) | 0.10 (0.04) | 0.24 | **0.002** | | | |
| LLFDI function UE | 0.08 (0.03) | 0.19 | **0.015** | | | |
| LLFDI function BLE | 0.09 (0.04) | 0.25 | **0.011** | | | |
| LLFDI function ALE | 0.10 (0.03) | 0.28 | **0.004** | | | |
| **Other potential adherence mediators** | | | | | | |
| Group allocation aLiFE versus eLiFE | −1.45 (1.01) | −0.14 | 0.153 | | | |
| Risk screening moderate category | −0.08 (0.02) | −0.03 | **0.006** | −0.06 (0.03) | −0.21 | **0.035** |

Multivariate linear regression (n=120, aLiFE and eLiFE participants).
Bold p<0.05
*Only variables retained in the final model presented.
†Known risk factors for functional decline.
‡Factors which influence adherence.
§Strength and balance deficits known to influence adherence.
ALE, advanced lower extremity; aLiFE, adapted Lifestyle-integrated Functional Exercise; BLE, basic lower extremity; CBMS, Community Balance and Mobility Scale; CES-D, Center for Epidemiological Studies Depression; eLiFE, enhanced Lifestyle-integrated Functional Exercise; LLFDI, Late-Life Function and Disability Index; MOCA, Montreal Cognitive Assessment; UE, upper extremity; VAS, Visual Analogue Scale.

## Treatment outcomes

Higher predicted adherence levels were significantly associated with improved LLFDI scores. In summary, one EARS point higher resulted in 0.67 point improvement on self-reported disability limitation (higher score indicates less perceived limitation) (F(3,102)=42.50,

**Table 4** Estimated association of adherence and group allocation at follow-up (aLiFE and eLiFE, n=120)

| | Unstandardised B | Coefficient SE | Stnβ | 95% CI | P value |
|---|---|---|---|---|---|
| **Primary outcome measures** | | | | | |
| LLFDI | | | | | |
| LLFDI disability frequency | | | | | |
| Baseline score | 0.677 | 0.065 | 0.716 | 0.549 to 0.80 | 0.000 |
| Group allocation | 0.872 | 0.553 | 0.105 | −0.226 to 1.970 | 0.118 |
| Predicted adherence level | 0.080 | 0.127 | 0.042 | −0.173 to 0.333 | 0.533 |
| LLFDI disability limitation (/100) | | | | | |
| Baseline score | 0.663 | 0.081 | 0.621 | 0.502 to 0.824 | 0.000 |
| Group allocation | −0.487 | 1.968 | −0.017 | −4.390 to 3.417 | 0.805 |
| Predicted adherence level | 0.663 | 0.081 | 0.621 | 0.346 to 2.393 | **0.009** |
| LLFDI function total components | | | | | |
| Baseline score | 0.779 | 0.064 | 0.760 | 0.652 to 0.906 | 0.000 |
| Group allocation | −1.354 | 1.260 | −0.057 | −3.854 to 1.146 | 0.285 |
| Predicted adherence level | 0.797 | 0.344 | 0.144 | 0.114 to 1.480 | **0.023** |
| LLFDI UE | | | | | |
| Baseline score | 0.624 | 0.074 | 0.623 | 0.478 to 0.771 | 0.000 |
| Group allocation | −0.667 | 1.694 | −0.027 | −4.028 to 2.693 | 0.694 |
| Predicted adherence level | 1.113 | 0.418 | 0.197 | 0.285 to 1.942 | **0.009** |
| LLFDI BLE | | | | | |
| Baseline score | 0.632 | 0.076 | 0.631 | 0.481 to 0.784 | 0.000 |
| Group allocation | −1.994 | 1.788 | −0.072 | −5.540 to 1.552 | 0.267 |
| Predicted adherence level | 1.304 | 0.486 | 0.204 | 0.339 to 2.268 | **0.009** |
| LLFDI ALE | | | | | |
| Baseline score | 0.868 | 0.066 | 0.780 | 0.736 to 0.999 | 0.000 |
| Group allocation | −1.934 | 1.763 | −0.057 | −5.431 to 1.562 | 0.275 |
| Predicted adherence level | 1.000 | 0.467 | 0.127 | 0.074 to 1.926 | **0.035** |
| Behavioural complexity | | | | | |
| Complexity, mean | | | | | |
| Baseline score | 0.178 | 0.076 | 0.235 | 0.028 to 0.329 | 0.021 |
| Group allocation | −0.012 | 0.019 | −0.063 | −0.049 to 0.025 | 0.525 |
| Predicted adherence level | 0.008 | 0.004 | 0.185 | −0.001 to 0.017 | 0.068 |
| Complexity, median (IQR) | | | | | |
| Baseline score | 0.209 | 0.076 | 0.270 | 0.057 to 0.360 | 0.008 |
| Group allocation | −0.007 | 0.019 | −0.035 | −0.045 to 0.031 | 0.726 |
| Predicted adherence level | 0.008 | 0.004 | 0.186 | −0.045 to 0.017 | 0.063 |
| **Secondary outcome measures** | | | | | |
| Cognition, affect and well-being | | | | | |
| MOCA score | | | | | |
| Baseline score | 0.590 | 0.090 | 0.558 | 0.411 to 0.769 | 0.000 |
| Group allocation | 0.190 | 0.328 | 0.049 | −0.461 to 0.842 | 0.563 |
| Predicted adherence level | −0.023 | 0.077 | −0.026 | −0.176 to 0.129 | 0.762 |
| CES-D score | | | | | |
| Baseline score | 0.426 | 0.087 | 0.444 | 0.253 to 0.598 | 0.000 |
| Group allocation | −0.586 | 0.903 | −0.048 | −2.377 to 1.205 | 0.518 |
| Predicted adherence level | −0.889 | 0.257 | −0.313 | −1.399 to −0.380 | **0.001** |

**Table 4** Continued

|  | Unstandardised B | Coefficient SE | Stnβ | 95% CI | P value |
|---|---|---|---|---|---|
| Short Form 12 |  |  |  |  |  |
| Baseline score | 0.500 | 0.102 | 0.467 | 0.297 to 0.703 | 0.000 |
| Group allocation | −0.990 | 0.517 | −0.158 | −2.017 to 0.037 | 0.059 |
| Predicated adherence level | 0.231 | 0.138 | 0.160 | −0.042 to 0.505 | 0.097 |
| Physical ability and mobility* |  |  |  |  |  |
| Usual gait speed (7 m walk) |  |  |  |  |  |
| Baseline score | 0.621 | 0.057 | 0.717 | 0.507 to 0.735 | 0.000 |
| Group allocation | 0.041 | 0.027 | 0.097 | −0.012 to 0.095 | 0.128 |
| Predicted adherence level | 0.014 | 0.007 | 0.141 | 0.001 to 0.027 | 0.036 |
| CBMS (/96) |  |  |  |  |  |
| Baseline score | 0.847 | 0.073 | 0.796 | 0.702 to 0.992 | 0.000 |
| Group allocation | 1.039 | 1.599 | 0.038 | −2.134 to 4.212 | 0.517 |
| Predicated adherence level | 0.206 | 0.431 | 0.033 | −0.648 to 1.061 | 0.633 |
| 8-Level Balance test |  |  |  |  |  |
| Baseline score | 0.467 | 0.116 | 0.396 | 0.237 to 0.697 | 0.000 |
| Group allocation | 0.254 | 0.213 | 0.111 | −0.168 to 0.677 | 0.235 |
| Predicted adherence level | 0.047 | 0.051 | 0.089 | −0.053 to 0.147 | 0.356 |

Bold p<0.05
*Strength and balance deficits known to influence adherence.
ALE, advanced lower extremity; aLiFE, adapted Lifestyle-integrated Functional Exercise; BLE, basic lower extremity; CBMS, Community Balance and Mobility Scale; CES-D, Center for Epidemiological Studies Depression; eLiFE, enhanced Lifestyle-integrated Functional Exercise; LLFDI, Late-Life Function and Disability Index; MOCA, Montreal Cognitive Assessment; UE, upper extremity; VAS, Visual Analogue Scale.

p<0.001). An improvement in all LLFDI function subcomponents was significantly associated with improved adherence. For each EARS point higher, upper extremity, basic lower extremity and advanced lower extremity function improved by 1.11 points ($F_{(3,102)}=35.90$, p<0.001), 1.30 points ($F_{(3,102)}=46.60$, p<0.001) and 1.0 points ($F_{(3,102)}=90.07$, p<0.001), respectively.

Behavioural complexity increased by 0.08 per point increase in EARS score; however, this change was not significant.

CES-D score was reduced by 0.89 points ($F_{(3,102)}=28.53$, p<0.001), while walking speed increased by 0.014 m/s ($F_{(3,102)}=50.28$, p<0.001) per point increase in EARS score.

## DISCUSSION

In this analysis, we have demonstrated that within this cohort, monthly reporting of adherence (prospective) correlated strongly with a single retrospective report of adherence (EARS) after a 6-month intervention period. We identified three main factors that influenced young older adults' adherence to their planned activity: the number of medications they were taking, their level of depression and their risk score of accelerated functional decline. Higher predicted adherence levels had a significant and clinically meaningful benefit on self-reported function (LLFDI) and greater behavioural complexity.

Further, predicted adherence was significantly associated with improved gait speed and lower depression scores. To our knowledge, this is the first analysis that examines the association of adherence to a lifestyle-integrated exercise programme with the intention to prevent functional decline in young older adults.

### Measuring adherence

To date, there is no consensus on how adherence should be measured. Of the existing approaches,[6] a non-standardised, arbitrary binary cut-off is commonly used to categorise participants into 'adherers' and 'non-adherers'. This approach has two main flaws. First, it fails to acknowledge that adherence is a complex construct of continuous quantity.[18] Second, dichotomising participants based on an arbitrary cut-off assumes that partial adherence has a nil treatment effect, when in fact some adherence may affect the dependent outcome measure.[33] Even not completing activities as often or as intended, any activity is still breaking up sedentary behaviour. Within this trial, we used monthly reporting of adherence, building on the gold standard for prospective fall data collection,[34] to eliminate recall bias due to delayed reporting. Further, we used the validated EARS questionnaire after the 6-month active intervention period. We found strong and highly significant correlation between the monthly (prospective) and the biannual (retrospective) responses, indicating that a single measurement timepoint could

suffice during the active intervention period. Further, there is a clear differentiation in mean EARS score and minimum scores among the three different adherence levels, supporting the opinion that 6 monthly EARS assessment can capture details of individuals' adherence levels without requiring monthly questionnaires and the associated potential reporting bias. Ultimately, the method chosen to measure adherence should be determined by its purpose.[6]

Monthly adherence reporting allows for tracking the change in adherence over time. From an intervention perspective, this could be used as a facilitator to identify adherence problems early on and to address potential barriers in a timely fashion. However, monthly adherence reports are also a form of intervention in themselves, as they acted as reminders to complete the planned activities.[14] Regular prompting has been shown to improve adherence,[35] and prompts in themselves are part of the BCT taxonomy.[36] Further, frequent reporting could also be considered burdensome on the participant, and a single retrospective questionnaire may be preferred. Within our young older population using EARS every 6 months seems sufficient to measure adherence accurately when the purpose of adherence documentation is to obtain an average score per participant, following a 6-month intervention period.

### Determinants of adherence

Adherence to the lifestyle-integrated activity programme varied among participants, with participants reporting on average 16 out of 24 points on EARS, indicating agreeable and positive intent to adhere. Such variability in adherence has repeatedly been reported for exercise interventions involving older adults[2] and is slightly higher than reported elsewhere.[17]

Previously, it has been demonstrated that external factors impact adherence behaviour such as location, frequency of exercising, level of supervision and delivery mode.[37] Also, there are numerous patient-specific, internal factors that directly or indirectly appear important for adherence behaviour.[10 11] Within our study population, the number of medications used, depression level and risk of accelerated functional decline were negatively associated with adherence. Increased medication use could be seen as a proxy for comorbidities, a barrier for adherence.[38] Surprisingly in our sample, the presence of arthritis or pain while walking was not an adherence barrier, as described in other studies.[39] However, reported pain levels were low, and bias from underenrolment of those with severe arthritis is possible.

While exercise is an effective intervention to manage depression,[40] depression adversely impacts adherence to exercise interventions, as shown here and in other studies.[41] Measuring participants' depression levels, an established adherence barrier, before commencing an intervention may help predict subsequent adherence and also allow for the tailoring of the intervention and support of participants accordingly.

The third factor which impacted adherence in this study was risk of accelerated functional decline. Those at high risk were limited by the burden of already deteriorating health, as shown previously,[11] while those at moderate risk may have understood the urgency to undertake preventative measures and were still capable of doing so.[42] Previous exercise programmes in participants with cardiovascular conditions have shown that education and awareness regarding one's health can have a positive effect on intervention adherence levels.[43] Participants within our trial were informed of their risk profile and had undergone detailed assessment, which highlighted their balance and strength deficits, and could have acted as a 'prompt' to adhere to the intervention.

When enrolling participants in an intervention programme, assessing physical and psychological adherence barriers can help tailor the intervention delivery to suit participants' preferences and needs.[41] This provides an opportunity to identify non-adherers early and target known barriers, for example, by applying BCTs to address psychological barriers, such as poor self-efficacy or poor motivation (or depression),[35] or by multidisciplinary interventions to manage pain and comorbidities better.

### Estimated association of adherence

Adherence to an intervention programme influences the intervention dose, which for PA interventions can lead to an improvement in physical outcome measures.[18 44] Predictive modelling showed that higher adherence levels were significantly associated with improved subjective health and function (LLFDI), as well as with improved behavioural complexity scores, gait speed and depressive symptoms. These results are in line with the previous reporting of balance training improving self-reported function in community-dwelling older adults,[45] as well as exercise improving depression levels[46] and increasing gait speed.[47] However, several other physical outcome measures remained unchanged, despite higher estimated adherence levels. These inconclusive findings are similar to those by Hughes et al,[3] who demonstrated that even interventions that focus specifically on improved adherence did not achieve meaningful change in clinical outcome measures. Future trials need to understand whom, within their patient population, will adhere and why; and which adherence barriers can be addressed and modified. Further work is also needed to better understand whether adhering to the dose, type or exercise intensity prescribed has the same or different impact on outcome.

The analysis presented here spans the first 6 months of the intervention period. Overall, the adherence rates were low, with an average EARS score of 16.75 points at 6 months in the eLiFE group and 15.29 in the aLiFE group. These results are in line with findings from other technology-supported trials, which have shown adherence rates to drop as time passes, with only 25% of participants still adhering at the end of a 48-week home-exercise programme.[48] The slightly higher adherence rates in eLiFE could be due to the functionalities available within the app, which included

daily reminders to complete the exercise, motivating messaging and alternative exercise suggestions.

Within the intervention period, participants in eLiFE and aLiFE received four or six home visits, respectively. Participants were encouraged to integrate their activities into everyday routines, without supervision. Although unsupervised activities have been shown to improve self-reported adherence,[49] this did not translate into an improvement in clinical outcome measures. A possible explanation is that while better adherence is reported, the quality, accuracy and intensity of the activity are not ideal when unsupervised, limiting its benefits. Further work incorporating technological measures of adherence is needed to document accurately the quality and fidelity of the activity undertaken in an unsupervised setting. Beyond documenting exercise execution, an Information and Communication Technology (ICT) platform could provide feedback to participants about how and what to improve during the intervention programme. Simultaneously, healthcare providers and researchers could access this information and use telecommunication to increase adherence, optimise exercise accuracy/quality and increase the likelihood of achieving an improvement in clinical outcomes.

### Strength and limitations

A strength of this study was that the activity programme offered to the participants was not detached from daily life but instead was integrated into existing routines. Providing an opportunity to be more active has been labelled a key element to achieving better adherence rates both with[50] and without[8] the use of technology. Within this trial, the intervention was personalised and adapted to an individual's capabilities, similar to approaches by other researchers who individualised interventions integrated into existing routines.[51 52] Within our study,[16] different BCTs were applied and combined to improve adherence, as this has been identified as an essential intervention component to promote sustainable changes in activity habits.[36] Future studies should evaluate technology-delivered methods which apply different BCTs to unravel the added benefit of both components.

During analysis, adherence was treated as a continuous variable, which fulfils the fifth assumption (exclusion restriction) in the so-called 'CACE analysis'.[29–31] Using the continuous EARS score to quantify adherence can capture the effect of proportionate adherence.

Several limitations of this study are acknowledged. The estimate of the association of adherence only allows us to determine if participation levels are associated with better outcomes, and it does not estimate the causal effects of the treatment since there is no comparison with a comparable group. As the intention-to-treat analysis showed no difference in effect between groups, a suitable adherer 'average causal effect analysis' was not possible. Two different treatment modes are compared based on adherence levels. It was not possible to assess whether adherence to eLiFE has greater implications on clinical outcomes than adherence to aLiFE. However, both intervention modes apply the same activity framework, and analysis to determine the estimated effect of adherence was defined a priori. Future work should consider further subgroup analysis to understand the adherence to different intervention methods better. Lastly, monthly measurement of adherence could have acted as a prompt to complete the planned activities. Within our study all participants were asked to report adherence regardless of group allocation; therefore, if adherence was influenced by monthly questioning, this was consistent for all participants.

### Implications for future research and clinical practice

Group allocation did not impact adherence in this study; however, research[53] has shown that adherence can be increased when patients can select the intervention mode. Offering patients a choice of intervention (patient preference) could impact adherence rates without compromising evidence-based interventions.[54]

To encourage higher rates of adherence, barriers such as depressive symptoms or polypharmacy could be addressed as part of a preintervention to enhance the ability to test new interventions.

In future trials, we should consider measuring baseline characteristics that affect adherence levels. This may include variables not generally considered for standard data collection, such as social support to support interventions, what exercise programmes have previously been explored and outcome expectations.[16] With improved baseline prediction of adherence and patient preference interventions, subsequent interventions can be more accurately assessed.

### CONCLUSION

PA adherence was associated with better lower extremity function and physical behavioural complexity. Barriers to adherence should be addressed preintervention to enhance intervention efficacy. Further research is needed to unravel the impact of BCTs embedded into technology-delivered activity interventions on adherence.

**Author affiliations**
[1]Department of Clinical Gerontology, Robert-Bosch-Krankenhaus GmbH, Stuttgart, Germany
[2]Department of Physiotherapy, Oslo Metropolitan University, Oslo, Norway
[3]Department of Neuromedicine and Movement Science, Norwegian University of Science and Technology, Trondheim, Norway
[4]School of Health Sciences, The University of Manchester, Manchester, UK
[5]Health & Care Policy, Age UK, London, UK
[6]Manchester Academic Health Science Centre, Manchester, UK
[7]Department of Human Movement Sciences, The University of Melbourne, Melbourne, Victoria, Australia
[8]Department of Electrical, Electronic and Information Engineering, University of Bologna, Bologna, Italy
[9]Ecole Polytechnique Federale de Lausanne, Lausanne, Switzerland
[10]Department of Human Movement Sciences, Vrije Universiteit Amsterdam, Amsterdam, The Netherlands
[11]Manchester University NHS Foundation Trust, Manchester, UK

**Acknowledgements** We thank all the participants in the trial, additional medical staff, assessors and instructors at the three clinical sites, and all partners in the consortium for their support, assistance and input throughout the different phases of the project.

**Contributors** KT, EB, HH-H, ABM, SM, KA, LC, MP, CT, BV, JLH and CB designed the research; ASM, KT, EB, HH-H, KG and AP-I conducted the research; ASM, KT, KG and AP-I analysed the data, performed the statistical analysis and interpreted the outcomes; ASM, KT and EB prepared the manuscript; ASM, KT, EB, BV, JLH, CT and CB were primarily responsible for the final content; all authors read and approved the final version of the manuscript. CB is responsible for the overal content as the guarantor.

**Funding** This work was supported by the European Union (EU)-funded project PreventIT (2016–2018, grant number 689238) responding to the Horizon 2020, Personalized Health and Care call PHC-21: Advancing Active and Healthy Aging with ICT: Early Risk Detection and Intervention. The EU was not actively responsible or involved in the study design, collection, management, analysis or interpretation of data. The writing of reports and the decision to submit for publication was not authorised by the EU.

**Competing interests** SM and LC own a share in the spin-off company of the University of Bologna, mHealth Technologies s.r.l. The remaining authors declare that the research was conducted in the absence of any commercial or financial relationships that could be construed as a potential conflict of interest.

**Patient and public involvement** Patients and/or the public were involved in the design, or conduct, reporting or dissemination plans of this research. Refer to the Methods section for further details.

**Patient consent for publication** Not applicable.

**Ethics approval** This study involves human participants and was approved by Trondheim, Norway (REK midt, 2016/1891), Stuttgart, Germany (registration number 770/2016BO1) and Amsterdam, The Netherlands (registration number 2016.539; Dutch Trial Registry NL59977.029.16). Participants gave informed consent to participate in the study before taking part.

**Provenance and peer review** Not commissioned; externally peer reviewed.

**Data availability statement** Data are available upon reasonable request. The datasets presented in this article are not readily available because the datasets for this study are not openly accessible to all but shared only between project partners due to ethical approval guidelines in the three clinical sites that conducted the multicentre trial, and cannot be shared until 10 years after the end of the project. Requests to access the datasets should be directed to kristin.taraldsen@ntnu.no.

**ORCID iDs**
A Stefanie Mikolaizak http://orcid.org/0000-0002-3732-3052
Kristin Taraldsen http://orcid.org/0000-0002-8466-5450
Elisabeth Boulton http://orcid.org/0000-0003-2791-8295
Katharina Gordt http://orcid.org/0000-0002-2709-1147
Andrea Britta Maier http://orcid.org/0000-0001-7206-1724
Sabato Mellone http://orcid.org/0000-0001-7688-0188
Helen Hawley-Hague http://orcid.org/0000-0002-2451-4482
Kamiar Aminian http://orcid.org/0000-0002-6582-5375
Lorenzo Chiari http://orcid.org/0000-0002-2318-4370
Anisoara Paraschiv-Ionescu http://orcid.org/0000-0002-2341-4383
Mirjam Pijnappels http://orcid.org/0000-0001-8416-2602
Chris Todd http://orcid.org/0000-0001-6645-4505
Beatrix Vereijken http://orcid.org/0000-0002-2231-8138
Jorunn L Helbostad http://orcid.org/0000-0003-0214-9290
Clemens Becker http://orcid.org/0000-0003-1624-8353

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
