## [Reviewer comments · BMJ Open]

ARTICLE DETAILS

TITLE (PROVISIONAL)	Impact of adherence to a Lifestyle-integrated Programme on physical function and behavioural complexity in young older adults at risk of functional decline: a multicentre RCT secondary analysis.
AUTHORS	Mikolaizak, A. Stefanie; Taraldsen, Kristin; Boulton, Elisabeth; Gordt, Katharina; Maier, Andrea; Mellone, Sabato; Hawley-Hague, Helen; Aminian, Kamiar; Chiari, Lorenzo; Paraschiv-Ionescu, Anisoara; Pijnappels, Mirjam; Todd, Chris; Vereijken, Beatrix; Helbostad, Jorunn; Becker, Clemens

VERSION 1 – REVIEW

REVIEWER	Susanne Finnegan University of Warwick, Warwick Clinical Trials Unit
REVIEW RETURNED	11-Aug-2021

GENERAL COMMENTS	Thank you for the opportunity to review this paper on adherence to the aLiFe and eLiFe programmes. Long-term adherence to interventions like this is such a challenge so this is an important piece of work to start to unpick what we might do to overcome these challenges. Abstract: an accurate overview of the study. Background: a good introduction to the issues with adherence to exercise interventions. Page 6 - Line 8: do you mean causal inference not interference? Methods: Page 9 - line 10: how many home visits is several and was this across the six months to help progress exercises or mainly at the beginning to teach the exercises - can you add a little more detail for replication purposes? Page 9 - line 52: it would be good to see an image/copy of what your adherence logs look like (in an appendix/supplementary information) to be able to picture what you have asked the participants to do to capture adherence. Page 10 - lines 47 - 58: your description of what the participants are asked to report do not match the reporting of adherence i.e. full adherence was summarized as "all or more than planned" but "all completed" is not an option to participants - do you mean "more than or as much as planned"? and does "not as much as planned" in the summary equate to "less than planned"? I think the wording just needs to match a little better for clarity. Results: this sections addresses the research question adequately. Discussion: Page 17 - line 29 - 33: This sentence about measuring baseline depression levels is a little confusing and may need re-writing. Page 17 - line 45 - 49: This sentence about the positive effects of education is also slightly confusing and could be re-written. Page 18 - lines 33 - 45: these are interesting findings and raise the
---

	question about different types of adherence i.e. even though people are adhering to the dose and type of exercise, are they actually achieving the intensity of exercise that is required to make changes to specific clinical outcomes. Therefore, are there different levels of adherence that need to be addressed? You touch on this later in the discussion and I think this is a really important point. Page 19 - line 3: could you just give a brief list of what these functionalities of the app are?
--	---

REVIEWER	Geeske Peeters Radboudumc, Geriatric Medicine
REVIEW RETURNED	25-Sep-2021

GENERAL COMMENTS	General comments: I commend the authors for discussing an important topic using a pragmatic approach that suits the context of lifestyle intervention trials. While it is generally acknowledged that adherence is an important element of intervention success, few studies investigate adherence beyond the reporting of (often poorly measured) adherence rates. This study does make that extra step by additionally examining determinants of adherence and associations of adherence with study endpoints. The study is conducted in people aged 60-70 years old; a group that is increasingly targeted in lifestyle and multicomponent interventions aiming to prevent health problems such as functional decline, falls, dementia, etc. Therefore, this study makes an important contribution to the field. Overall this is a well designed study and clearly written paper, but some details would benefit from further clarification. Specific comments: -p. 3, line 15: "We report an a priori subgroup analysis to 1) compare adherence measures, ..." After reading this, my interpretation of the first sub-aim was that subgroups were compared on adherence measures. After reading the background (p. 8, line 5), I understand that the first sub-aim was to examine whether prospectively measured adherence gives the same results as retrospectively measured adherence. Please rephrase the aim in the abstract to avoid confusion. -p. 3, line 17: "... the impact of adherence on outcome measures". Maybe add "of adherence" for clarity. -p. 4, line 6: Based on the abstract, it seems that the study did not look at barriers, specifically. Consider removing this sentence on barriers from the conclusion in the abstract. -p. 7, lines 24 and 49-58: This study is described as an 'a-priory subgroup analyses' of an RCT. It seems, however, that the analyses were done in the total intervention group, not a subgroup. Could you please clarify if, or how, the participants whose data are included in the current analyses form a subgroup of the total intervention group? Are the in/exclusion criteria provided here (lines 49-58) for the original study (if so, were there additional in/exclusion criteria for the subgroup analyses) or for this subgroup analyses (if so, what were the original in/exclusion criteria for the RCT)? -p. 9, lines 26-47: To facilitate interpretation of the results, could you please add the range of scores or reference values for the behavioural complexity outcome? What is a good/poor score? -p. 11, line 47-51/Table 3: Why were not all variables with statistically significant univariate associations included in the multivariable models? According to the text on page 11, it seems that all significant invariable associations were included in the
---

	multivariable models, but Table 3 shows only significant associations in the multivariable model. As a result, it is unclear which variables were actually included in the multivariable model (were non-significant variables kept in the model or not?). -p. 11, lines 54 and 60: From the info in line 60, I understand that not observed (raw) EARS scores were used, but that 'predicted' adherence levels were derived from linear regression results. Why was this method used? What did the regression models look like? To facilitate interpretation, could you please provide the range of scores and reference values (good/poor scores) for the predicted values? Note that using the term 'predicted adherence' for the exposure in the main analyses is a bit confusing, as the term 'predicted' is commonly used for the outcome of an analysis. -p. 12, lines 35-49/Table 2: I'm not sure how useful the correlation between the monthly adherence scores (reduced to three categories) and the 6-month EARS score is to demonstrate that these two measures measure the same thing. It may be more informative to add mean (and min/max) EARS scores for each category of the monthly adherence in Table 2. One would expect a step increase in mean EARS scores for each higher category of monthly adherence. -p. 13, line 12: Strictly taken, it is unclear if these determinants 'influenced' adherence, but you can say that they were 'associated with' poorer adherence. This part of the analyses is based on observational data (not an experiment) and thus causality cannot be assumed. -p. 16, lines 10 and 15/Table 3: number of medications is associated with adherence and proposed that this may be a proxy for comorbidity, which is a barrier for adherence. From Table 1, I understand that information on comorbidities is available. Why was this not included as a potential determinant in Table 3? -p. 19, line 19 - study limitations: Could the monthly measurement of adherence somehow have affected the adherence and/or the retrospective reporting of adherence by making participants more aware of their behaviour? -p. 19, line 19 - study limitations: many tests were done. How likely is it that multiple testing may be led to type I error? -p. 20: A conclusion paragraph is missing.
--	---

VERSION 1 – AUTHOR RESPONSE

Reviewer 1		
Page 6 - Line 8: do you mean causal inference not interference?	This has been updated to read: 'causal inference' .	Background
Page 9 - line 10: how many home visits is several and was this across the six months to help progress exercises or mainly at the beginning to teach the exercises - can you add a little more detail for replication purposes?	More detail has been added and now reads "Participants received between one and six home visits, depending on group allocation, from an exercise physiologist/scientist or physiotherapist across six months during which the exercise program was taught and participants were supported to progress their exercises independently." Also we have added reference to the	Page 9, Intervention

	main publication, which lists the complete intervention in detail allowing replication. “. The full interventions details have been published (12).	
Page 9 - line 52: it would be good to see an image/copy of what your adherence logs look like (in an appendix/supplementary information) to be able to picture what you have asked the participants to do to capture adherence	The adherence was measured via a single question, asked in the form of a seven-option questionnaire. The following has been added to the manuscript to further clarify “ Every month, participants were asked to report, on a single-page questionnaire by ticking one of seven options, whether they had completed their planned activities.... ”.	Monthly reporting of adherence, page 10
Page 10 - lines 47 - 58: your description of what the participants are asked to report do not match the reporting of adherence i.e. full adherence was summarized as "all or more than planned" but "all completed" is not an option to participants - do you mean "more than or as much as planned"? and does "not as much as planned" in the summary equate to "less than planned"? I think the wording just needs to match a little better for clarity.	Further, to the response the above question, the wording has been aligned within the manuscript, to clarify how the patients reported adherence corresponded with their overall adherence rating.	Monthly reporting of adherence, page 10
Page 17 - line 29 - 33: This sentence about measuring baseline depression levels is a little confusing and may need re-writing.	We have re-written this paragraph, which accidentally was missing part of a sentence. It now reads as follows “ Measuring participants’ depression levels, an established adherence barrier, before commencing an intervention may help predict subsequent adherence and also allow for the tailoring of the intervention and support of participants accordingly. ”	Discussion, determinants of adherence
Page 17 - line 45 - 49: This sentence about the positive effects of education is also slightly confusing and could be re-written	The sentence regarding the effect of education has been rewritten: “ Previous exercise programmes in participants with cardiovascular conditions have shown that education	Discussion, determinants of adherence

	and awareness regarding one's health can have a positive effect on intervention adherence levels"	
Page 18 - lines 33 - 45: these are interesting findings and raise the question about different types of adherence i.e. even though people are adhering to the dose and type of exercise, are they actually achieving the intensity of exercise that is required to make changes to specific clinical outcomes. Therefore, are there different levels of adherence that need to be addressed? You touch on this later in the discussion and I think this is a really important point.	As it is further discussed later in the Discussion, we have only added a short sentence highlighting the need for further work. "Further work is also needed to better understand whether adhering to the dose, type or exercise intensity prescribed has the same or different impact on outcome"	Discussion, Estimated association of adherence
Page 19 - line 3: could you just give a brief list of what these functionalities of the app are?	We have added some exemplary functionalities "which included daily reminders to complete the exercise, motivating messaging and alternative exercise suggestions" .	Discussion, Estimated association of adherence
Reviewer 2		
p. 3, line 15: "We report an a priori subgroup analysis to 1) compare adherence measures, ..." After reading this, my interpretation of the first sub-aim was that subgroups were compared on adherence measures. After reading the background (p. 8, line 5), I understand that the first sub-aim was to examine whether prospectively measured adherence gives the same results as retrospectively measured adherence. Please rephrase the aim in the abstract to avoid confusion.	The abstract has been updated with the objective now stating "Objectives: 1) compare adherence measures; 2) identify determinants of adherence, and 3) assess the impact on outcome measures of a lifestyle-integrated programme.	Abstract
p. 3, line 17: "... the impact of adherence on outcome measures". Maybe add "of adherence" for clarity.	We have amended the abstract's conclusion	Abstract
p. 4, line 6: Based on the abstract, it seems that the study did not look at barriers, specifically. Consider removing this sentence on barriers	As we identified determinants of adherence, which if present lower adherence level, we believe these could be seen as barriers. We have	

from the conclusion in the abstract.	therefore decided to leave the text unchanged.	
p. 7, lines 24 and 49-58: This study is described as an 'a-priory subgroup analyses' of an RCT. It seems, however, that the analyses were done in the total intervention group, not a subgroup. Could you please clarify if, or how, the participants whose data are included in the current analyses form a subgroup of the total intervention group? Are the in/exclusion criteria provided here (lines 49-58) for the original study (if so, were there additional in/exclusion criteria for the subgroup analyses) or for this subgroup analyses (if so, what were the original in/exclusion criteria for the RCT)?	The a-priory analysis is based on adherence levels, but includes all intervention group participants. We have clarified this by removing the term subgroup, as this was misleading.	Methods
p. 9, lines 26-47: To facilitate interpretation of the results, could you please add the range of scores or reference values for the behavioural complexity outcome? What is a good/poor score?	The definition of normative values for any PA metrics is a challenging topic since many factors such as age, gender, health status, life context, etc., have to be considered. Therefore, in order to identify the range of the developed PA behavioural complexity metric we analyzed data recorded with a similar sensor configuration (single accelerometer on sacrum/trunk) in samples of various populations, as follows: frail elderly, well-functioning older adults without and with mild concern about falling, medical staff working in intensive care hospital unit, and farmers working in a rural area in African country. The value of behavioural complexity metric ranged from a minimal value of 0.1 for very impaired frail elderly to a maximal value of around 0.7 for highly active subjects (e.g., African farmers). Published data for well-functioning older adults indicated a value of 0.40 ± 0.07 (mean \pm std) for those fully confident about a fear of falling, and	Methods

	0.30±0.06 for those active but less confident [Paraschiv-Ionescu et al., 2018]: Paraschiv-Ionescu, Anisoara, et al. "Concern about falling and complexity of free-living physical activity patterns in well-functioning older adults." Gerontology 64.6 (2018): 603-611. The following sentence has been added to the methods section: "The value of behavioural complexity metric ranges from a minimal value of 0.1 for very impaired frail elderly to a maximal value of around 0.7 for highly active subjects. For context, published data for well-functioning older adults indicated a value of 0.40±0.07 (mean±std) for fully confident older adults without a fear of falling, and 0.30±0.06 for those active but less confident in their ability."	
p. 11, line 47-51/Table 3: Why were not all variables with statistically significant univariate associations included in the multivariable models? According to the text on page 11, it seems that all significant invariable associations were included in the multivariable models, but Table 3 shows only significant associations in the multivariable model. As a result, it is unclear which variables were actually included in the multivariable model (were non-significant variables kept in the model or not?).	All nine significant univariate associations were included in the multivariate model. However, only three significant variables were retained in the multivariate linear regression model. We have extended the statistical analysis paragraph to include that the MLR was a backwards MLR model. "Variables identified during univariate regression (p<0.05) were subsequently entered into a backward (elimination) multivariate linear regression model to determine the best set of explanatory variables."	Method, Statistical Analysis
-p. 11, lines 54 and 60: From the info in line 60, I understand that not observed (raw) EARS scores were used, but that 'predicted' adherence levels were derived from linear regression results. Why was this method used? What did the regression models look like? To facilitate interpretation, could you please provide the range of scores	Self-reported, retrospective adherence scores as measured using EARS can range from 0-24, with higher scores indicating better adherence. This is explained in the methods section as well as Table 2. Predicted adherence measured using EARS has the same range and interpretation of results.	Methods, statistical analysis

and reference values (good/poor scores) for the predicted values? Note that using the term 'predicted adherence' for the exposure in the main analyses is a bit confusing, as the term 'predicted' is commonly used for the outcome of an analysis.

In order to understand the effect of adherence the preferred statistical method is to undertake a complier average causal effect (CACE) analysis. However, as mentioned in the limitation section of this manuscript, the intent-to-treat analysis needs to be significant for a CACE analysis to be suitable. In lieu of a 'proper' CACE analysis, we still applied the same assumptions to our analysis, which includes using data collected pre-randomization. The EARS scores however represent the adherence as reported at 6 months, which is the time point at which this questionnaire is correctly completed. While it is the correct time to be assessing adherence, it results in post-randomization scores and therefore this variable is excluded from being used within an average causal effect analysis.

Backward multiple linear regression was used to determine which model of baseline variables best fits to explain subsequent EARS score. In order to use EARS scores, we had to predict participants' EARS scores based on outcome measures assessed prior randomization. Therefore we have to clarify that it is 'predicted' adherence, as is based on their baseline variables rather than their true adherence.

We appreciate that the term 'predicted' is used differently than usual, however it would be incorrect to not label the 'predicted' EARS scores as such, when trying to determine how adherence levels are associated with outcome measures.

The statistical analysis paragraph has been expanded to address the reviewer's comment: *In lieu of a*

	suitable complier average causal effect (CACE) analysis, which was not possible given the intention-to-treat analysis showed no difference in effect between groups, multiple linear regression was performed while still honouring the CACE assumptions [26]	
p. 12, lines 35-49/Table 2: I'm not sure how useful the correlation between the monthly adherence scores (reduced to three categories) and the 6-month EARS score is to demonstrate that these two measures measure the same thing. It may be more informative to add mean (and min/max) EARS scores for each category of the monthly adherence in Table 2. One would expect a step increase in mean EARS scores for each higher category of monthly adherence.	The mean as well as min/max range of EARS score per adherence status has been added to Table 2. The results section has been amended to include a sentence regarding these findings. "The mean EARS scores increased with increased adherence, as measured on a monthly basis, with non-adherers scoring as low as 4 out of 24 points and full adherers having a mean of 20 out of 24 on their EARS score." We have expanded the discussion accordingly which now includes: "Further, there is a clear differentiation in mean EARS score and minimum scores amongst the three different adherence levels, supporting the opinion that six-monthly EARS assessment can capture details of individuals' adherence levels without requiring monthly questionnaires and the associated potential reporting bias."	Table 2; Results;
p. 13, line 12: Strictly taken, it is unclear if these determinants 'influenced' adherence, but you can say that they were 'associated with' poorer adherence. This part of the analyses is based on observational data (not an experiment) and thus causality cannot be assumed.	We agree that causality cannot be assumed and have insured that we state that the relationship is an 'association' rather than an influence. "Key factors which were anticipated to positively or negatively be associated with participants' adherence..."	Methods, determinants of adherence
p. 16, lines 10 and 15/Table 3: number of medications is associated with adherence and proposed that this may be a proxy for comorbidity, which is a barrier for adherence. From Table 1, I	There are several reasons why the variables 'medication use' was within the regression model, rather than comorbidities. 1) Given the number of	Statistical analysis

understand that information on comorbidities is available. Why was this not included as a potential determinant in Table 3?	observations, we were restricted in the number of variables we could use.  2) Many variables measure the same 'outcome' in different ways. To avoid violating the collinearity assumption, we had to make an educated decision, which variable we would include from each 'domain'. 3) Comorbidity information was obtained from participants using a non-validated questionnaire, as this was appropriate for the original PreventIT trial. Medication use information was obtained by reviewing all medication participants were taking. We felt this was a more objective measure and was therefore more suited as a variable within the regression. Using both variables was not possible due to reason 2). 4) Lastly the number of comorbidities was low and therefore the data skewed within this population of younger older adults, making it a suboptimal variable to use in analysis. We have clarified our variable selection within the methods section "To ensure the collinearity assumption was met, potential variables were required to be measuring/assessing uniquely different outcomes"	
p. 19, line 19 - study limitations: Could the monthly measurement of adherence somehow have affected the adherence and/or the retrospective reporting of adherence by making participants more aware of their behaviour?	This is a very good point and we have added it to the limitations of this manuscript and analysis. "Lastly, monthly measurement of adherence could have acted as a prompt to complete the planned activities. Within our study all participants were asked to report adherence regardless of group allocation, therefore if adherence was influenced by monthly questioning this was consistent for all participants."	Discussion, limitations

p. 19, line 19 - study limitations: many tests were done. How likely is it that multiple testing may be led to type I error?	As mentioned above, in the analysis we selected only the recommended number of variables based on the number of observations available to avoid Type I errors. “a restricted number of variables was selected determined by the number of available observations to reduce the risk of type I error.”	Statistical analysis
p. 20: A conclusion paragraph is missing	A conclusion paragraph has been added “Physical activity adherence was associated with better lower extremity function and physical behavioural complexity. Barriers to adherence should be addressed pre-intervention, to enhance intervention efficacy. Further research is needed to unravel the impact of behaviour change techniques embedded into technology-delivered activity interventions on adherence”	Conclusion

VERSION 2 – REVIEW

REVIEWER	Susanne Finnegan University of Warwick, Warwick Clinical Trials Unit
REVIEW RETURNED	07-Dec-2021
GENERAL COMMENTS	Thank you for responding to all of my comments and making the appropriate changes. I feel that your manuscript should now be accepted for publication.